# Allometry reveals trade-offs between Bergmann's and Allen's rules, and different avian adaptive strategies for thermoregulation

Arkadiusz Fröhlich [1]✉, Dorota Kotowska[1], Rafał Martyka[1] & Matthew R. E. Symonds [2]

Animals tend to decrease in body size (Bergmann's rule) and elongate appendages (Allen's rule) in warm climates. However, it is unknown whether these patterns depend on each other or constitute independent responses to the thermal environment. Here, based on a global phylogenetic comparative analysis across 99.7% of the world's bird species, we show that the way in which the relative length of unfeathered appendages co-varies with temperature depends on body size and vice versa. First, the larger the body, the greater the increase in beak length with temperature. Second, the temperature-based increase in tarsus length is apparent only in larger birds, whereas in smaller birds, tarsus length decreases with temperature. Third, body size and the length of beak and tarsus interact with each other to predict the species' environmental temperature. These findings suggest that the animals' body size and shape are products of an evolutionary compromise that reflects distinct alternative thermoregulatory adaptations.

The tremendous diversity of physical forms shown by life on earth[1] has long fascinated biologists and prompted calls for explanations. Two of the oldest attempts to explain this phenomenon are Bergmann's (1847)[2] and Allen's (1877)[3] rules, which link variation in animal body size and shape with climate. Bergmann's rule states that body size tends to decrease toward the equator, as small body size (which produces a higher surface-area-to-volume ratio) results in more effective heat exchange with the ambient environment and hence is advantageous in warm conditions. Conversely, large body size (lower surface-area-to-volume ratio) effectively reduces heat loss, thus is favorable in cold conditions[2]. Allen's rule states that animals with longer appendages (hence higher surface-area-to-volume ratio) are more likely to be found in hot climates, where shedding excess body heat may be needed. In turn, shorter appendages (lower surface-area-to-volume ratio) are more likely found in cold conditions, where effective retention of warmth is required[3]. Notably, both Bergmann's and Allen's rules invoke the argument that the body's surface-area-to-volume ratio plays a key role in thermoregulation, and has been the subject of natural selection as animals adapted to their thermal environments. Changes in the

ambient temperature over evolutionary time either through environmental changes in situ[4,5] or when species colonized new environments[6–8] have therefore driven diversity in animal body size (Bergmann's rule) and shape (Allen's rule) as they adapted to these new thermal environments.

For the last 150 years, an array of evidence has accumulated that Bergmann's and Allen's rules reliably link phenotypic variation with geographic and temporal distribution across populations[9–13] and species[14–18]. Accordingly, body sizes tend to decrease[9,10,14,15] and relative appendage sizes tend to increase[13,16–18] with the environmental temperature. Since the middle of the 20th century, the global climate has been experiencing warming decade-to-decade, resulting in observations that animals have been shrinking in size and/or elongating appendages[11,12,19–21], which suggests that Bergmann's and Allen's rules describe widespread and fundamental macro-evolutionary patterns. However, we are far from understanding how body size and shape evolve simultaneously in response to thermal conditions as consensus as to how Bergmann's and Allen's patterns interact has not been yet established. Few studies have explicitly examined[13,21,22] whether

[1]Institute of Nature Conservation, Polish Academy of Sciences, Mickiewicza 33, Kraków 31-120, Poland. [2]Centre for Integrative Ecology, School of Life and Environmental Sciences, Deakin University, Burwood, Victoria 3125, Australia. ✉e-mail: frohlich@iop.krakow.pl

geographic patterns in body size and shape occurs simultaneously across multiple species, although a recent study within 26 Australian shorebird species[22] indicated support for both rules, but did not indicate any specific association between the two. The failures to identify complementarity in both rules may be a consequence of the analytical approach, or the limited taxonomic and/or geographic coverage of previous studies[13,21,22]. To better address the question of how the two rules are complete we need to capture broad gradients of temperatures and phenotypes.

If both Bergmann's and Allen's rule are true, an animal's phenotype has two ways to adapt to novel climates—first through the shift in body size, and second through changes in the size of appendages. These two elements of adaptation to climate may both be selected to change in a complementary manner. Alternatively, or one or other may be selected for in a compromise due to evolutionary or physical constraints, as in addition to thermoregulation, the phenotype has to meet functional requirements, such as foraging or metabolic efficiency[23,24].

The concept of an evolutionary compromise, or 'trade-off', between body size and shape is not new. Indeed, Allen explicitly embedded his rule in the context of Bergmann's rule, proposing a scenario where animal lineages could maintain their optimum body size (or even show converse Bergmann's patterns) under increasing temperatures through changes in the size of appendages over evolutionary time[3]. According to this scenario, the size of appendages should increase with temperature more steeply in large-bodied organisms and increase less (or possibly remain constant) in small-bodied organisms. This is because having a small body may be sufficient to deal with hot temperatures, without the need to change body shape. By contrast, large-bodied organisms (according to Bergmann's rule) may easily overheat at higher ambient temperatures, hence should be under stronger selection to evolve elongated appendages to disperse excess heat. Simultaneously, body size should decrease with temperature more steeply in organisms with small appendages, and less so in organisms with large appendages, because large-appendaged organisms (according to Allen's rule) already effectively exchange the heat with the ambient environment, thus their body sizes might be under weaker selection for thermoregulation in warm climates. Thus, the climate-dependent development (and maintenance) of either large or small body size may depend on the size of appendages and vice versa. If the above is true, historical shifts in body size (Bergmann's rule) and the size of appendages (Allen's rule) may represent two distinct evolutionary ways to cope with thermoregulatory changes and we need not necessarily predict that both will hold simultaneously. This may also in part explain why concordance with Bergmann's rule or Allen's rule is not universal across lineages[13,24,25].

Allen's scenarios have implications for allometry[26]—i.e. the relationship between the absolute size of appendage and body size[27–29]. If Allen was right, and body size determines how the relative size of appendages increases with temperature (a stronger Allen's rule relationship in larger animals), then the allometry of appendage size should vary across temperature gradients, with the increase in the absolute size of appendages (against body size) expected to be steeper in warm climates and milder in cold climates. To test this hypothesis, we may evaluate if the interaction of temperature and body size predicts the absolute size of appendages[30], allowing us to determine how the slope of Allen's rule pattern varies across various settings of body size, and simultaneously, whether and how allometry of appendage size varies across a temperature gradient.

A compromise between Bergmann's and Allen's rule, if it exists, could be reflected in many ways. First, when animals adapt to warmer climates, they should either become smaller (Bergmann's rule) or their appendages should elongate relative to body size (Allen's rule). Here, we would expect there to be an interaction between body size and the relative size of appendages when predicting the ambient temperature experienced by animals (which reflects the thermal preferences of those species), with implications for how their phenotypes adapt to different climates. Second, if temperature influences body size (Bergmann's rule), then the temperature should influence also the relative size of appendages (Allen's rule), assessed as the residuals of the regression of appendage size against body size, when body size is corrected by the geographic temperature. Whereas, if the compromise hypothesis is false, the temperature will better predict the size of appendages relative to raw body size (not-accounting for environmental context). This may be tested by integrating allometry, Bergmann's rule, and Allen's rule in single causal model, which releases the assumption of mutual independence between body size and appendage length[31].

Here, we address this issue with a phylogenetic comparative analysis of nearly all bird species (9962 species ~ 99.7% of global diversity). We use phylogenetically-informed[32] linear regression models[33] to examine how the temperature[34] experienced by species within their geographic ranges[35] co-varies with their body size[36] (Bergmann's rule) and the relative sizes of unfeathered appendages: the beak and the tarsus[1] (Allen's rule), which have been linked to thermoregulation[16,37,38]. To disentangle the question of whether these rules describe complementary or alternative strategies to cope with thermoregulatory demands, we examine how body size interacts with temperature when predicting the size of appendages and (simultaneously) how temperature influences the interspecific allometry of appendage length. We then investigate how body size and appendage length interact with each other to predict the environmental temperature experienced by the species within its geographic range ('environmental temperature' hereafter) to ask how the phenotype adapts to different climates. Finally, we integrate Bergmann's rule, Allen's rule, and allometry in various causal models[39,40] to better understand how the temperature gradients affect the evolutionary shifts in phenotype and how the phenotype is linked to evolutionary adaptations to different climates. We test all of these hypotheses using various temperature measures (ranging from lower, through average, to upper temperatures occurring annually within species range) and we control for avian migratory habits (which notably influence experienced temperatures[41] and phenotypes[42]). We also examine whether our findings differ between species of different range sizes, to exclude possible confounds of climatic- and phenotypic-variability within continental and cosmopolitan species[8,22]. In addition, we use different measures of relative appendage sizes (residuals, ratios and principal components) and multiple phylogenetic trees to reduce possible artefacts from particular choices of measures or evolutionary reconstructions.

## Results and discussion
### Bergmann's rule
Variation in avian body size has arisen through millions of years of evolution[43], and our data reflects this by showing that log body mass is strongly predicted by phylogeny (Supplementary Table 1). Yet, avian body size also shows large geographical variation (Fig. 1a), and our analysis provides strong support for Bergmann's rule across the global community of birds. Phylogenetic linear mixed models indicated that the temperature variables explain from 9.0% to 11.8% of the variance in log-transformed body size (estimated with r-squared; Fig. 1b). These models are substantially better supported than the null model and the model with latitude alone (Fig. 1b), suggesting that the observed geographical pattern is linked to thermoregulation. All of these temperature models indicate that temperature negatively correlates with body size (Fig. 1c and Supplementary Fig. 1), as predicted by Bergmann's rule.

### Allometry of appendages
Allen's hypothesis[3] implies that the length of animal's appendages varies with temperature in relative (not absolute) terms, thus when

asking how the appendage length vary across temperature gradient, we always need to control for body size. Phylogenetic log-log regression models revealed that body mass explains 72.7% and 72.5% of variance in beak and tarsus length (estimated with r-squared of models shown in Fig. 2a and Supplementary Fig. 3a), respectively, confirming that the evolution of absolute avian appendage size is substantially constrained by body size. These null allometric models predict that log-transformed beak length (Fig. 2a) and tarsus length (Fig. 3a) scale with log-transformed body mass in a linear manner:

$$\log_e(\text{Beak Length}) = 1.4345 + 0.3362 \log_e\text{Body Mass} \qquad (1)$$

$$\log_e(\text{Tarsus Length}) = 2.1141 + 0.2883 \log_e\textit{Body}\text{ Mass} \qquad (2)$$

the normalized formulas of which give us the logarithmic equations:

$$\text{Beak Length} = 4.1975\, \text{Body Mass}^{0.3362} \qquad (3)$$

$$\text{Tarsus Length} = 8.2821\, \text{Body Mass}^{0.2883} \qquad (4)$$

Because these allometric plots (Figs. 2a and 3a) relate the length of the appendage (one dimensional linear measure) to the body mass (three-dimensional volumetric measure) it means that the size of appendages would scale isometrically (proportionally)

with the body size if the allometric coefficient was 0.3333. Thus, beak length equals to body mass to a power of 0.3362 means that the beak elongates almost exactly proportionally with body size. However, tarsus length equals to body mass to a power of 0.2883 means that the extent to which tarsus elongates with body size is slightly more pronounced in smaller species and weaker in larger species.

These allometric relationships have implications for how we interpret subsequent patterns. For example, consider a species that experience a temporal increase in temperature, or invades a warmer climate. Then, if only Bergmann's rule is operating (and in the absence of other confounds), a gradual decrease in body size should result in a proportional decrease in absolute beak length, and a gradually larger decrease in absolute tarsus length. Conversely, if species follow only Allen's rule (and not follow Bergmann's rule), then the increase in beak length should be similar between larger and smaller species, while the increase in tarsus length should be weaker in larger species and stronger in smaller species. Thus, Allen's assumption that the increase in the ratio of body width to body length is steeper in larger species[3], should not be a direct effect of the allometric rules, as appendages tend to increase proportionally with body size (beak) or increase milder at larger body sizes (tarsus).

## Allen's rule

After excluding the effect of allometry, relative beak length is still tightly associated with phylogeny (Supplementary Table 1), while showing an impressive geographic variation (Fig. 2b). Our phylogenetic analysis

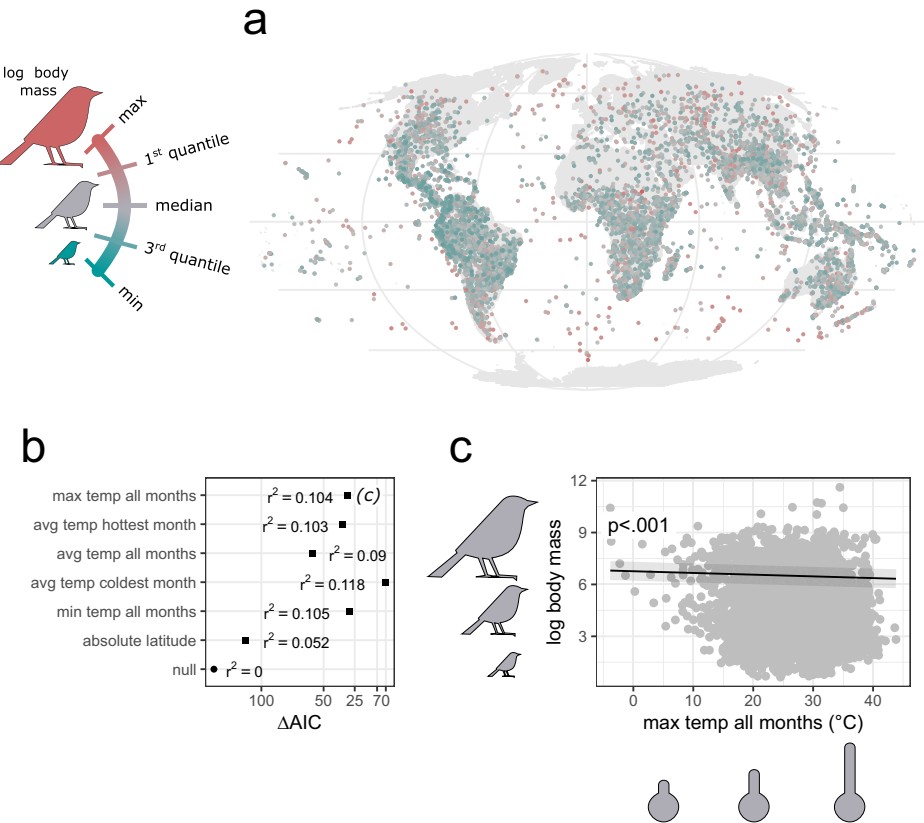

**Fig. 1 | Global test of Bergmann's rule across 9962 (99.7%) avian species.** Distribution of log-transformed body mass across species geographic ranges, shown as their geometric centroids (**a**). Model selection procedure for predicting log body mass (**b**), with six temperature measures assessed within species geographic ranges, as sole fixed effects; AIC—Akaike Information Criterion, r²—coefficient of determination. An exemplar Bergmann's model (**c**), showing decreasing body size with max temperature of all months; see Supplementary Fig. 1 for surrogate models based on the other temperature measures (evaluated in **b**). The shaded area around the trend line is simple shading to facilitate reading. The p values refer to the significance of temperature effect and whether it differs from zero as derived from a two-tailed test. The results were obtained with phylogenetic linear regression by *phylolm* models on a single maximum clade credibility phylogenetic tree.

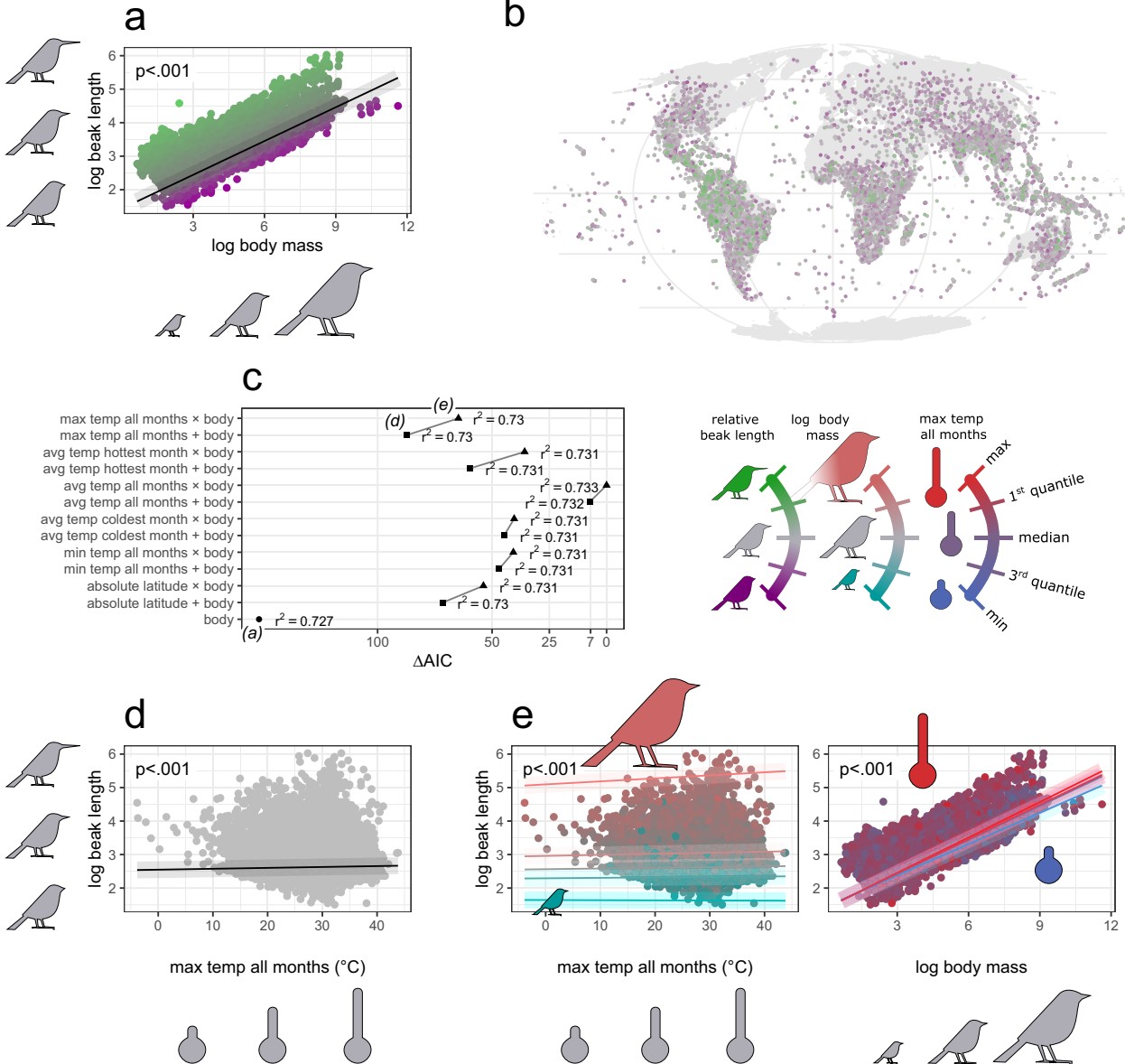

**Fig. 2 | Global test of Allen's rule on avian beak length across 9962 (99.7%) bird species.** The null allometric model (**a**) used to scale the absolute (log-transformed) beak length with log body size, the residuals from which were used as the relative beak length. Distribution of relative beak length across species geographic ranges (**b**). Model selection procedure for predicting log beak length (**c**), involving models with log body mass and either of six temperature measures within species geographic ranges included as fixed and interaction terms; AIC−Akaike Information Criterion, r²−coefficient of determination. An exemplar Allen's model (**d**) showing increasing beak length with max temperature of all months, while controlling for body size as fixed term. An exemplar model with interaction of body size and max temperature of all months (**e**) illustrating how Allen's rule operates across steeping quantiles of body size (left) and how allometry varies across quantiles of temperature (right). See Supplementary Fig. 2 for surrogate models based on the other temperature measures (evaluated in **c**). The *p* values refer to the significance of model's fixed (**d**) or interaction terms (**e**) derived from two-tailed tests. The shaded area around the trend line is simple shading to facilitate reading. The results were obtained with phylogenetic linear regression by *phylolm* models on a single maximum clade credibility phylogenetic tree.

concurs with an array of existing studies[12,16,17,44,45] that found that the length of avian beak follows Allen's rule, and is a general pattern across birds as a whole. Among our models predicting beak length, those with temperature variables among fixed terms are more informative than the null allometric model, where log body mass (allometry) is put as sole predictor (Fig. 2c). Most of the temperature variables also predict beak length better than latitude (Fig. 2c), again confirming the thermoregulatory basis of the observed pattern. Each of the temperature variables are positively associated with longer beaks (Fig. 2d and Supplementary Fig. 2a), which remains in agreement with Allen's rule.

Some studies have reported the ambiguous[46] or very weak[16] Allen's pattern for avian legs. While relative tarsus length is also well conserved in avian phylogeny (Supplementary Table 1) and shows a

high geographic variation (Fig. 3b), surprisingly, our global phylogenetic analysis indicates that avian tarsus length follows the inverse of Allen's rule. Among models explaining tarsus length, those with temperature variables are better than the null allometric model (Fig. 3c). However, these models indicate a negative correlation− thereby shorter tarsi are associated with warmer temperatures (Fig. 3d).

**Allen's vs Bergmann's rule in allometry**

Our analyses support the hypothesis that the way in which avian appendages size varies across temperature regimes, depends on body size and vice versa. First, among models of beak length, those with an interaction of body size and the temperature consistently

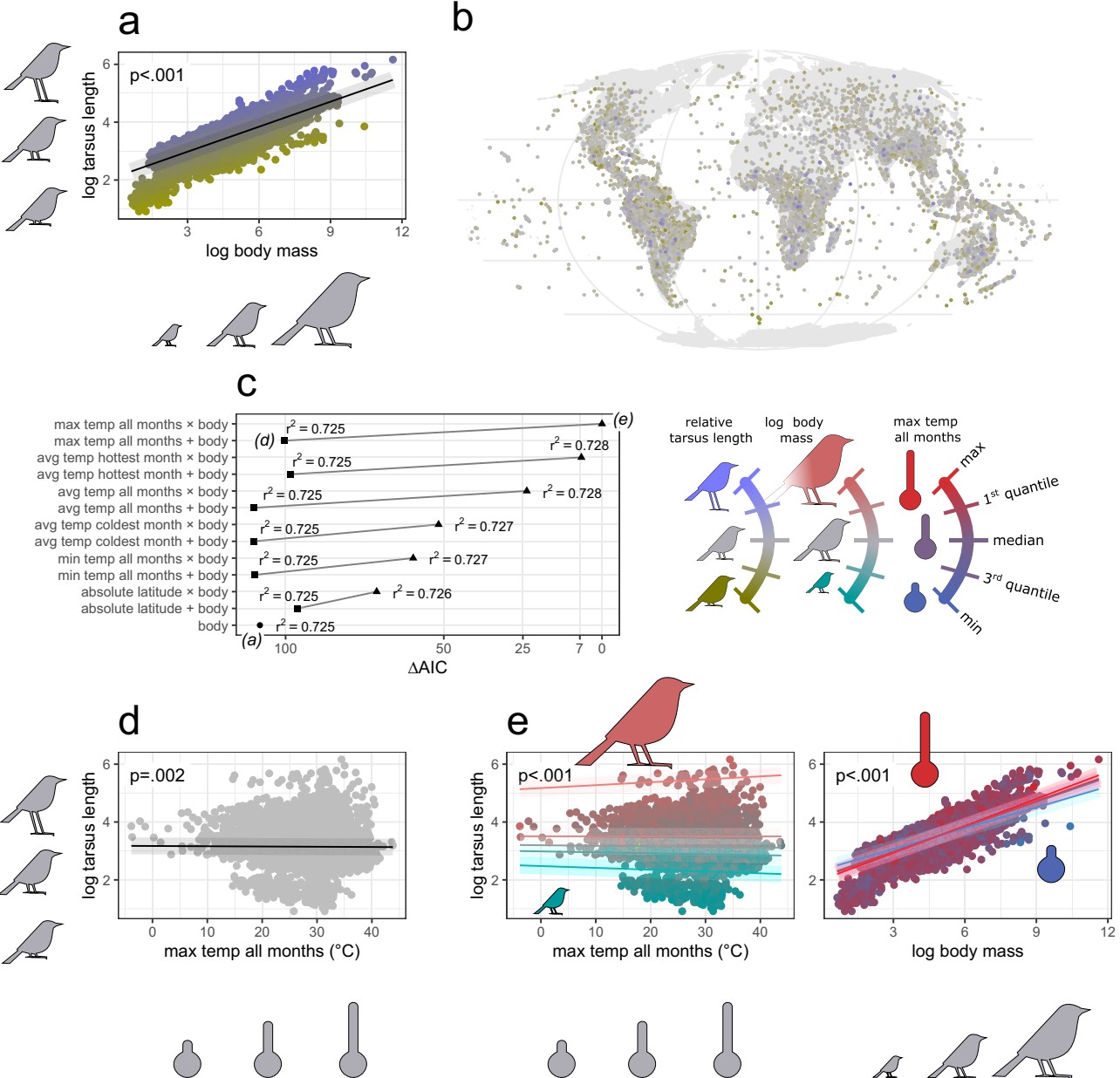

**Fig. 3 | Global test of Allen's rule on avian tarsus length across 9962 (99.7%) bird species.** The null allometric model (**a**) used to scale the absolute (log-transformed) tarsus length with log body size, the residuals from which were used as the relative tarsus length. Distribution of relative tarsus length across species geographic ranges (**b**). Model selection procedure for predicting log tarsus length (**c**), involving models with log body mass and either of six temperature measures within species geographic ranges included as fixed and interaction terms; AIC−Akaike Information Criterion, r²−coefficient of determination. An exemplar Allen's model (**d**) showing decreasing tarsus length with max temperature of all months, while controlling for body size as fixed term. An exemplar model with interaction of body size and max temperature of all months (**e**) illustrates how Allen's rule operates across steeping quantiles of body size (left) and how allometry varies across steeping quantiles of temperature (right). See Supplementary Fig. 3 for surrogate models based on the other temperature measures (evaluated in **c**). The *p* values refer to the significance of model's fixed (**d**) or interaction terms (**e**) derived from two-tailed tests. The shaded area around the trend line is simple shading to facilitate reading. The results were obtained with phylogenetic linear regression by *phylolm* models on a single maximum clade credibility phylogenetic tree.

perform better than models without that interaction (Fig. 2c). The interaction of temperature and body size loads positively on beak length, indicating that larger-bodied species show stronger increases in beak size with temperature (Fig. 2e, left plot). Notably, beak length does not co-vary with temperature in the smallest birds (Fig. 2e, left plot), which is in agreement with Allen's speculations[3] that being smaller reduces the need to develop elongated appendages in hot climates, as effective heat exchange is already enabled through small body size (according to Bergmann's rule). The positive interaction between body size and temperature also indicates

that the higher the temperature, the steeper the allometric relationship between beak size and body size (Fig. 2e, right plot), meaning that in warmer climates beak size increases more strongly with body size than in colder climates, exactly as Allen hypothesized.

An interaction between body size and temperature is also consistently supported in models of tarsus length (Fig. 3c). This interaction has strong positive effect on tarsus length, thereby reversing the trend by which tarsus shortens with temperature (Fig. 3e, left plot). This means that despite the overall decrease of tarsus size with

temperature in smaller birds (the inverse of Allen's rule), the opposite is true for larger birds that show increasing tarsus size with temperature (Fig. 3e, left plot). The interaction holds regardless of the temperature measure examined (Supplementary Fig. 3b, upper row), even if those previously did not co-vary with tarsus length when included as simple independent term with body size (Supplementary Fig. 3a). The case of larger birds thus fits Allen's rule, and agrees with Allen's further speculations[3] that appendages are more likely to increase in larger- than in smaller-bodied animals. However, Allen did not predict the possibility of shortening appendages toward hot temperatures as seen in small birds. Given the extent of our sampling, the effect of shortening tarsi toward the equator in small-bodied species is presumably not an artefact, but relies on yet unknown mechanisms (possibly unrelated to thermoregulation). Nevertheless, if there is an evolutionary pressure to develop a smaller tarsus in hot climates, the increased thermoregulatory needs of larger-bodied species possibly overwhelm this selective process. This may be because large species acquire higher heat loads when the ambient temperature is hot, hence necessitating the development of longer legs as cooling organs. As with beak size, the interaction also indicates substantial changes in allometry, with much more millimeters of tarsus per each gram of body in warm conditions compared to cold conditions (Fig. 3e, right plot).

Our analyses also support the mirror scenario, that the extent to which body size decreases with temperature (Bergmann's rule) depends on the length of appendage. In models predicting body size, the temperature does not interact with relative beak length (Supplementary Fig. 4a), but interacts with tarsus length (Supplementary Fig. 4b). This interaction indicates that the strongest shrinkage in body size with temperature occurs in shorter-legged birds, while in longer-legged birds body size increases with temperature (inverse Bergmann's rule). This again supports Allen's speculations that variation in body shape allows birds to evolve body sizes less restricted (or even unrestricted) to environmental temperature. Thus, the results support the theory that Bergmann's and Allen's rules are two distinct, albeit analogous strategies to deal with thermoregulation.

## Allen's vs Bergmann's rule in climatic adaptations

Our analysis shows that the interactions of body size (Bergmann's rule), beak length and tarsus length (Allen's rule) predict the thermal environment across birds (e.g. the max temperatures of all months across species ranges, Fig. 4a). As with body size and shape, the temperatures experienced by species within their geographic ranges are finely conserved in the avian phylogeny (Supplementary Table 1), suggesting that thermal preferences of avian species have been established through evolutionary history. Evolution of these preferences then occurred when temperature changes affected their native environments (thus causing extinctions or adaptations), or when birds invaded novel environments (thus adapting to newly-encountered climates). Log-transformed body mass, relative beak length and tarsus length clearly predict the species ambient temperature (Fig. 4b), suggesting that the phenotype changes as animals adapt to suit different climates. However, of particular note is that the addition of an interaction between body size and relative beak length substantially improves model performance (Fig. 4b). This interaction shows that for longer-beaked birds, temperature associations are unrelated to body size, but the shorter the beak, the more pronounced is the shrinking in body size in warmer temperatures (Fig. 4c, left plot). In the case of smaller-bodied birds, the adaptation to different temperatures is independent of beak length, but with larger birds, the adaptation to warmer temperatures is more likely associated with elongated beaks (Fig. 4c, right plot). These results indicate that living in warmer temperatures tends to be associated either with smaller body size (Bergmann's rule) or longer beak (Allen's rule), rather than both rules simultaneously, thus again supporting the hypothesis of an

evolutionary compromise between shifts in body size and shape as alternative adaptations to thermal environment.

The interaction of body size and relative tarsus length also substantially improves the model predicting ambient temperature of the species (Fig. 4b). This interaction indicates that living in warmer climates is associated with smaller body size (Bergmann's rule) only in shorter-legged birds, while in longer-legged birds the environmental temperature increases with body size (inverse Bergmann's rule; Fig. 4d, left plot). Simultaneously, the avian environmental temperature increases with tarsus length (Allen's rule) only in larger species, while the opposite is true for smaller species (Fig. 4d, right plot). This suggests that larger-legged avian lineages may be resistant to Bergmann's rule and become larger when habituating to warm climates, while shorter- and average-legged birds become smaller with temperature, as predicted by Bergmann's rule. These findings again converge with Allen's speculations on trade-off in the evolution of body size and appendage length in relation to temperature.

We found that the length of the two different appendages−beak and tarsus−show independent evolutionary patterns (Fig. 4e). The environmental temperature of a species increases with beak length independently from tarsus length, and decreases with tarsus length independently from beak length (Fig. 4e). These outcomes reject the possibility of an evolutionary compromise in climatic adaptation of two types of appendages, at least when we do not control for body size (Bergmann's rule) as additional type of climatic adaptation.

Finally, the model with a three-way interaction between body size, relative beak and tarsus length predicting temperature performs the best among all considered candidate models (Fig. 4b) and this interaction is statistically significant (Fig. 4f), suggesting that evolutionary adaptation to novel climates depends on various configurations of body size, beak, and tarsus length. This model indicates various Bergmann's rule slopes across different settings of body shape (Fig. 4f, top-left). Namely, the steepest decrease in environmental temperature with body size (i.e. strongest Bergmann's rule) is observed in smaller-billed and smaller-legged birds (Fig. 4f, top-left, brown trend line), whereas in longer-billed and shorter-legged birds (Fig. 4f, top-left, green trend line) body size is not associated with environmental temperature. This model also indicates that in shorter-billed, longer-legged birds (Fig. 4f, top-left, purple trend line) body size increases across temperature gradient (inverse Bergmann's pattern). This thus strengthens the support for Allen's theory that having bodies with elongated appendages may enable species to circumvent or even reverse Bergmann's pattern; whereas compact bodies are more prone to decrease in size with temperature in order to deal with overheating in warm climates. Counteracting this argument, however, is that longer-billed and longer-legged birds show (moderate) typical Bergmann's pattern (Fig. 4f, leftmost plot, bluish trend line).

The three-way interaction model also shows other mixtures of expected and unexpected results. For example, the strongest increase in environmental temperature with beak length occurs in larger-bodied and shorter-legged birds (Fig. 4f, top-right plot, orange trend line), which clearly suggests a trade-off in evolution of body size and beak length and a similar trade-off in the evolution of the two types of appendages, presumably reflecting different adaptive responses for thermoregulation. However, a similar increase in beak length also occurs in tiny-bodied and longer-legged birds (Fig. 4f, top-right plot, blue trend line), which stands in contrast to this trade-off hypothesis. Likewise, the steepest increase in environmental temperature with tarsus length (Allen's rule) occurs in larger-bodied and shorter-billed birds (Fig. 4f, bottom plot, pink trend line), again suggesting a compromise scenario, with elongated tarsus evolving as thermoregulatory organ to compensate for insufficient heat exchange due to large body and small beak. It also suggests that, in large birds, having a short beak in hot climates requires longer tarsi (Fig. 4f, top-right, pinkish and orange trend lines) and vice versa (Fig. 4f, bottom plot, rose and

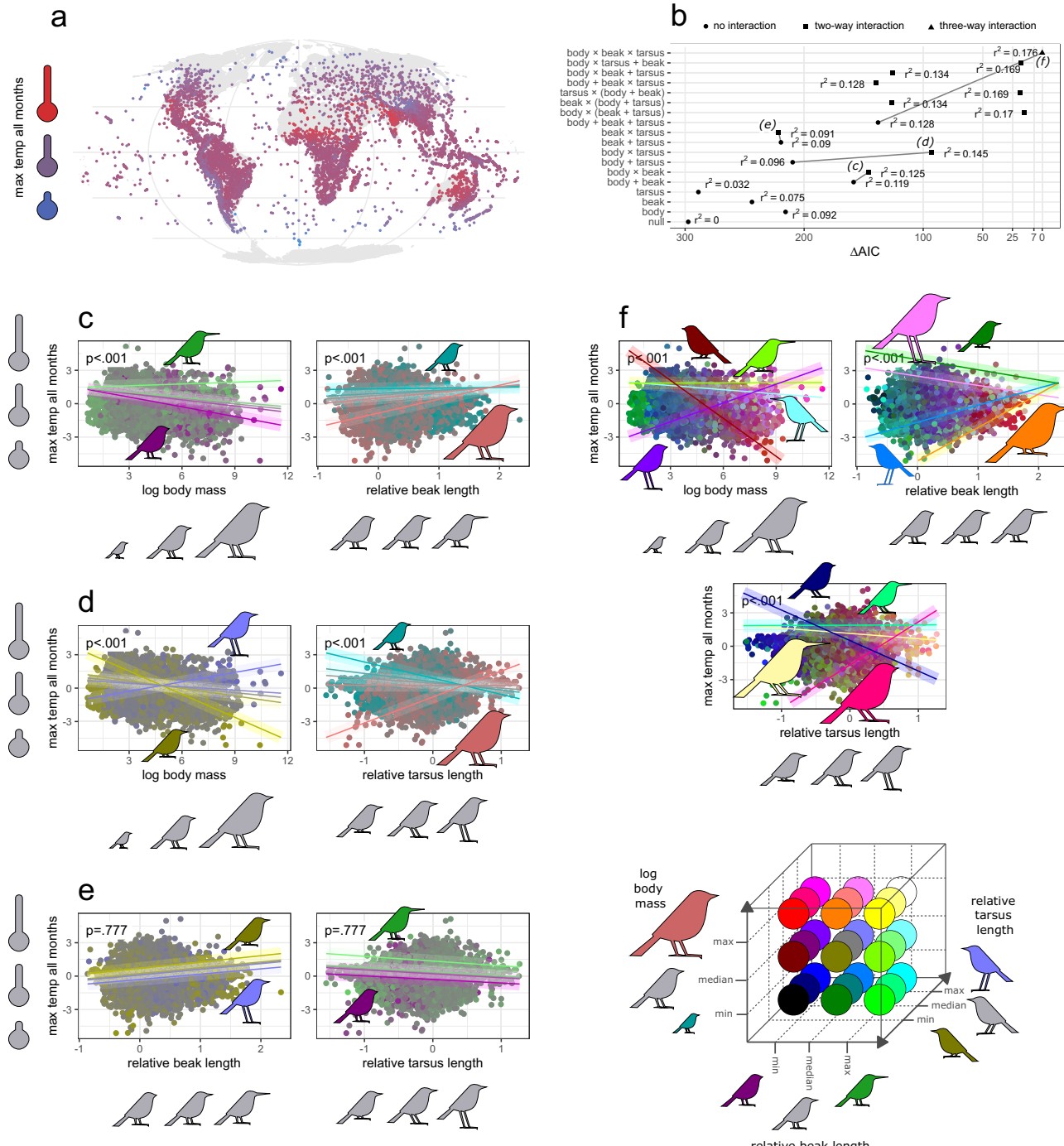

**Fig. 4 | Global test for avian adaptation to maximum temperature across all months by shifts in body size (Bergmann's rule) and appendage size (Allen's rule) across 9962 (99.7%) avian species.** Distribution of environmental temperature across species geographic ranges (**a**). Model selection procedure for predicting max temperature all months (**b**), involving models with different combinations of log body mass, relative beak and tarsus length as fixed and interaction terms; AIC−Akaike Information Criterion, r²−coefficient of determination. Exemplar models with two-way interaction of body size and relative beak length (**c**) or tarsus length (**d**) illustrate how Bergmann's rule operate across steeping quantiles of relative appendage length (left plots) and how Allen's rule operate across steeping quantiles of body size (right plots). An exemplar model with two-way interaction of relative beak length and tarsus length (**e**) illustrates how Allen's rule based on the relative length of one appendage operates across steeping quantiles of the relative length of second appendage. An exemplar model with three-way interaction of log body mass, relative beak and tarsus length (**f**) illustrates how shifts in body size and two measures of body shape depend on each other when animals adapt to novel climates; the trend lines indicate relationships between $y$ and $x_1$ (axes) across combinations of min and max values of $x_2$ and $x_3$ (colors); see also Supplementary Fig. 5 for more detailed visualization of the model **f**. The $p$ values refer to significance of two-way (**c**−**e**) and three-way (**f**) interaction terms derived from two-tailed tests. The shaded area around the trend line is simple shading to facilitate reading. Obtained with phylogenetic linear regression by *phylolm* models on a single maximum clade credibility phylogenetic tree.

yellowish trend lines), indicating that in large species, the summarized length of two types of appendages is important for thermoregulation. However, by contrast, it seems that in small bodied species, beak and tarsi length evolved in a correlated way (Fig. 4f, top-right and bottom plots, green and blue trend lines) across environmental temperature (occurrences in warmer temperatures are associated with simultaneously both longer beaks and tarsi, or else simultaneously shorter beaks and tarsi). This may indicate a general tendency to correlated evolution of relative beak and tarsus lengths, perhaps for functional reasons, e.g. longer beaks may allow long-legged birds to explore substrate more efficiently, as longer necks also do[47].

**Allen's vs Bergmann's rules in causal models**

Our hypothesis consequently holds within phylogenetic path analysis, where the best causal models integrate Bergmann's and Allen's rules to explain both the size of avian appendages (Fig. 5a) and the avian thermal environment (here, maximum temperature across all months) (Fig. 5b). The best model predicting beak and tarsus length includes the causal effect of temperature on body size (Bergmann's rule) and then body size on beak and tarsus length (allometry), as well as the direct effect of temperature on the size of appendages (Allen's rule) (Fig. 5a). This joint Bergmann's and Allen's model is substantially better than the model assuming that temperature does not affect body size before scaling for the length of appendages (Fig. 5a, Allen's rule only). The combined Bergmann's and Allen's model is also better than one assuming no direct effect of temperature on appendages (Fig. 5a, Bergmann's rule only). This again indicates that how the length of avian appendages co-varies with the ambient temperature partially depends on how avian body size co-varies with temperature, yielding results aligned with the trade-off hypothesis. This notably argues against the possibility that the increase in the length of appendages (relative to body size) with temperature is an artefact of decreased body sizes at hot temperatures (see[26]). However, interestingly, the model including only Allen's rule (and allometry) explains the length of appendages with similar accuracy to the model with only Bergmann's rule (Fig. 5a).

The best model predicting the temperature associations includes the indirect effect of body size on the length of appendages (allometry), and then the length of appendages on temperature (Allen's rule), as well as the direct effect of body size on temperature (Bergmann's rule) (Fig. 5b). These results again demonstrate that how the temperature varies across species ranges depends on both the size of body and appendages, suggesting that Bergmann's and Allen's rules describe two distinct evolutionary ways to cope with thermoregulation. Moreover, the similar performance of Allen's model compared to Bergmann's model (Fig. 5b) again suggests that shifts in the animal's body size and shape represent roughly equally influential in the evolution of adaptations to novel climates.

**Excluding possible confounding factors**

To ensure the reliability of our findings, first we show that when explaining the phenotype (Supplementary Figs. 2b and 3b), or the temperature within species geographic ranges (Supplementary Fig. 6), the main results remain consistent whichever of the five temperature measures is included. Second, despite the fact that the relationships with relative length of appendages and the experienced temperatures are strongest in resident birds, followed by partial- and full- migrants, our results still hold when accounting for these three categories of avian migratory habits (Supplementary Fig. 7); and the compromise scenario remains similar in each of these groups independently (Supplementary Figs. 8–10). It aligns with previous studies[22,46], which found that ecogeographical rules are valid regardless of variation in avian migratory habits. However, it is worth to notice that the most prominent trade-offs are found in resident species (in case of explaining environmental temperature, see Supplementary Fig. 10) or in partial migrants (in case of explaining beak

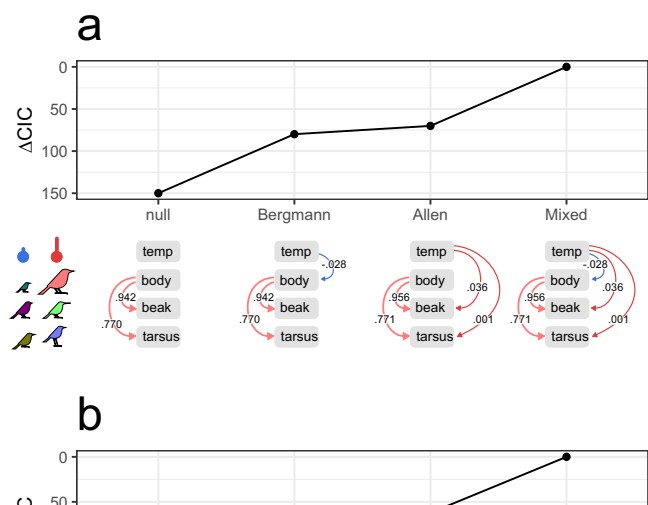

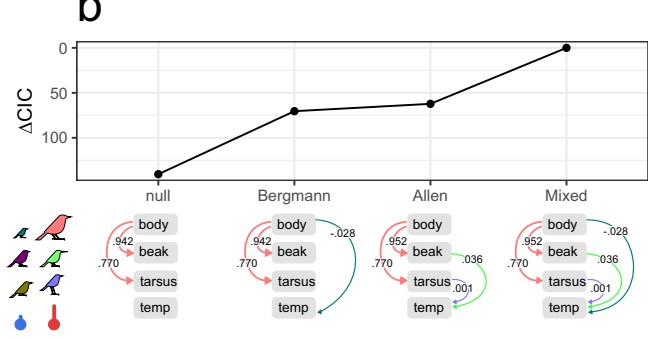

**Fig. 5 | Phylogenetic path analysis with responses of the length of avian appendages (beak and tarsus) (a) and the maximum temperature of all months within species range (b) across 9962 species (99.7% of global community).** In both cases model candidates include different combinations of allometry (relationship between the length of appendages and body size), Bergmann's rule (the relationship between body size and the temperature) and Allen's rule (relationship between the length of appendages and temperature). ΔCIC−delta C statistic Information Criterion. The results were obtained with phylogenetic path analysis by using *phylopath* models on a single maximum clade credibility phylogenetic tree and scaled covariates (mean = 0 ± 1 SD) to compare their effect sizes (see numbers on path diagrams).

length, see Supplementary Fig. 8). Third, the trade-offs in thermoregulatory strategies also hold after controlling for geographic range size (Supplementary Fig. 11) and remains quantitatively (Supplementary Figs. 12–13) or qualitatively (Supplementary Fig. 14) stable across the gradient of endemic-cosmopolitan species. Thus, even if ecogeographical rules operate within widespread species (across distanced populations, as well documented[9–13]), this does not appear to influences the results of our cross-species analysis. Fourth, the predictions of temperature within species geographic ranges are also not specific to the way by which we account for the allometry of appendages (by using residual appendage length). Parallel analyses with ratios of appendage length to body mass (Supplementary Fig. 15) or principal components of all phenotypic traits (Supplementary Fig. 16) give qualitatively similar outcomes. Fifth, we also show that results of both phylogenetic regression (Supplementary Fig. 17) and phylogenetic path models (Supplementary Fig. 18) remain consistently valid across 100 randomly chosen phylogenetic trees[32], mitigating concerns regarding phylogenetic uncertainty influencing our results.

Notably, there is a wider list of important ecological factors constraining or favoring variation in body size and shape, e.g. tropic levels or foraging techniques[21,23,47,48], although they are also themselves constrained by phylogeny to some extent, which we control for. Nevertheless, we believe that it is likely that these constraints influenced (or were influenced by) the Bergmann-Allen trade-off. Understanding of this issue would benefit from a deeper dive into the

relationship between climatic, phenotypic and ecological variation across animals.

## Our findings in the context of eco-evolutionary processes driven by climate

To the best of our knowledge, this study is the largest (taxonomically and geographically) simultaneous test of ecogeographical rules and it provides a first empirical evidence for a trade-off in the evolution of body size (Bergmann's rule[2]) and the size of appendages (Allen's rule[3]) across global temperature gradients. Our results confirm what Allen[3] speculated−the larger the body, the stronger the increase in appendage size with temperature; and the larger the appendages, the milder the decrease in body size with temperature. Thus, the evolution of body size under temperature regimes likely depends on the size of appendages and, on the other hand, the extent to which temperature drives the size of appendages depends on body size. This means that these two thermoregulatory adaptations are not independent of each other, but the phenotype has at least two ways to adapt to novel climates, i.e. by the shifts in body size or the shifts in the size of appendages (or both to a lesser extent).

The evolution of appendages (e.g. avian beaks[49]) was a dynamic process believed to overtake the changes in body size across evolutionary time[50]. Our analyses do not indicate, however, that shifts in body size have been more frequent than shifts in appendage size (or vice versa), at least not because of thermoregulation. Rather, they indicate that shifts in body size and shape are intertwined through avian evolutionary history, agreeing with the theory that animals select the most convenient strategy of thermoregulation to maintain functional traits of its phenotype. For functional reasons animal lineages tend to increase in body size over evolutionary time (Cope's rule[43]), thus it is not surprising that strategies allowing species to maintain/develop larger bodies (i.e. over-increase in appendage size) are to be expected evolutionarily. On the other hand, some lineages may be constrained in appendage size (e.g. to forage[21,47] or communicate[23] effectively), hence those may favor the shifts in body size to reconcile optimal thermoregulation with a desired functionality.

We found that the compromise in thermoregulatory strategies may also involve two distinct types of appendages, here beak versus tarsus. However, this is true only for larger-bodied species (see Fig. 4f top-right and bottom plots, trends for large bodies), that are more likely to acquire higher heat loads in warmer environments, thus the summarized size of many appendages may be for them crucial to disperse heat loads. Both beak and legs have been confirmed to act as key regions of heat transfer on the avian body[37,38,51,52], thus both may be sensitive to thermal conditions when body size is too large to deal alone with too hot temperatures. Yet, in small-bodied species both appendages seem to evolve in concerted way across temperature gradients, and this may be in a way that conforms with Allen's rule or not (see Fig. 4f, top-right and bottom plots, trends for small bodies), indicating that the small body ensures good temperature exchange in hot climates, thus the evolution of appendages in these species may be correlated, but independent of thermoregulatory selection pressures.

It is worthwhile emphasizing that apart from shifts in body size and shape, many other elements combine to help birds meet their thermoregulatory requirements[53], e.g. through variation in insulation (feathers)[54], coloration[55,56] metabolism[57], blood circulation[58] or behavior[59−61]. Extrapolating our results, these thermoregulatory strategies might also co-evolve under a trade-off to ensure optimal thermoregulation along with desired functionality. This is presumably a reason for the relatively low performance of our models; e.g. physical phenotype explains up to only 20% of the variance in ambient temperature (Fig. 4b, upper model), therefore unexplained variance must be attributed to other thermoregulatory strategies.

In this study, we demonstrate that Allen's rule may be attributed to the varying allometric functions across temperature gradients. Although logical and argued elsewhere[26], it has never been addressed by any empirical research. Our findings clearly indicate the importance of considering body mass as both a fixed and interaction term in studies of Allen's rule, but also might suggest that ambient temperature should be included in other allometric studies of animals' morphology. That said, temperature explains very little of the variance in the size of appendages compared to body size (Figs. 2c and 3c), thus thermal conditions are unlikely to be a very crucial confounding factor for allometry in comparative analyses.

In this study, we empirically confirm for the first time an evolutionary compromise theory that was first proposed almost 150 years ago[3]– the evolution of body size and appendages are two distinct and interacting ways to cope with thermoregulation. This may explain why many studies fail to detect Allen's or Bergmann's rules independently which has led to questioning of the generality of these ecogeographical patterns[13,24,25]. Here, our findings suggest that Bergmann's and Allen's rules should not necessarily be considered in isolation. We believe that these thermoregulatory strategies might intertwine through the evolutionary history of animals, as the evolution of phenotype possibly interacts to confound ecogeographical rules to evolve functional traits. This explanation also highlights the diverse mechanisms that animals may employ to expand across the world's multiple environments. It also raises the speculation that with observed and future anticipated warming of Earth's climate, we should expect mainly large animals to elongate in appendages, while mainly compact-bodied animals to shrink in size.

## Methods

### Phenotype

Body size (in grams), beak length (measured from the culmen to tip; in millimeters) and tarsus length (in millimeters) were retrieved from the *AVONET database*[1]. In this database, most of the body mass values come from Dunning's *Handbook of Avian Body Masses*[36]. All of the phenotype variables were log-transformed to achieve normal distribution and to enable allometric comparisons.

The database captured formidable variation in avian phenotypes. The median avian body mass was 35.5 g (e.g. White-browed woodswallow *Artamus superciliosus*), and ranged from 1.9 g (Peruvian sheartail *Thaumastura cora*) to 111 kg (Common ostrich *Struthio camelus*). After excluding the effect of allometry (see color gradients on log-log phylogenetic linear regression plot in Figs. 2a and 3a), the relative appendage lengths still present an impressive range of variation in the extent to which they are either larger or smaller than expected from body mass. The relative beak length ranged from −0.86 in the great dusky swift *Cypseloides senex* (which has an almost invisible beak), through 0.31 (median) found e.g. in the cinereous conebill *Conirostrum cinereum*, to 2.33 in the sword-billed hummingbird *Ensifera ensifera* (in which the beak accounts for approximately the 50% of the body length). The relative tarsus length ranged from −1.56 in rufous hummingbirds *Selasphorus rufus* (where the tarsus is one of the shortest bones in the body), through −0.01 (median) found e.g. in the black-whiskered vireo *Vireo altiloquus*, to 1.27 in the black-necked stork *Ephippiorhynchus asiaticus* (in which tarsus is one of the longest bones).

### Temperature

Climatic selection and constraint on the evolution of the desired phenotype may imply several scenarios, depending on which periods are most critical for thermoregulatory performance[62]. Some scenarios assume that phenotype is shaped by extreme temperature events (e.g. either the warmest or the coldest days or months), which cause severe mortality of organisms that can easily overheat or overcool during these critical timeframes[16,18,45,62]. Alternative scenarios assume that the phenotype is selected by the average temperature across year, as

animals spend less time on cooling or heating, and thereby performs better in foraging or reproduction. We therefore retrieved both average, upper and lower monthly temperatures measured within species ranges to test our hypotheses under these alternative scenarios.

We obtained temperature data for each species from spatial analyses within the 'sf' (version 1.0-8)[63] and 'raster' (version 3.5-15)[64] R packages, using global raster layers of temperatures available in World Clim database (version 2.1)[34]. These rasters (Tmin, Tavg and Tmax; see below) had a resolution of 30" and were consist of monthly averages from a period of 58 years (1960-2018). The temperature metrics have been calculated within polygons of species ranges available in form of multi-polygon vector layers extracted from the *BirdLife International database* (version 2020.1)[35].

We first excluded polygons identified as uncertain species presence, uncertain season of presence, non-native presence or species extinct in a region, leaving us with 9962 species (out of 9,993 species) with complete geographic data. Second, having polygons with only a certain, native and extant species presence we grouped them by the species (according to the phylogenetic taxonomy[32]) and the season of presence (either breeding season, winter or year-round presence) and then we aggregated them to obtain single polygons specific to species and season (Supplementary Fig. 19). Third, using breeding and year-round species ranges, where species live at hotter period of the year; we calculated their zonal means of monthly temperature maximums (Tmax) and took the largest monthly value for each species (maximum temperature of all months). We also calculated their zonal means of monthly temperature averages (Tavg) and took the largest monthly value for each species (average temperature of hottest month). Fourth, we analogously used winter and year-round species ranges, where species live at colder period of the year. Then, we calculated their zonal means of monthly temperature minimum (Tmin) and took the lowest monthly value for each species (minimum temperature of all months). We also calculated their zonal means of monthly temperature averages (Tavg) and took the lowest monthly value for each species (average temperature of coldest month). Fifth, we took all (breeding, winter and year-round) species ranges and we calculated their zonal means of monthly temperature averages (Tavg) and averaged all monthly values to obtain average temperature of all months for each species. We also retrieved absolute latitude from the centroids of the above species ranges (breeding, winter and year-round, summarized to a single polygon per species), which described a simple geographic variation across species. Sixth, the obtained temperature measures (minimum temperature of all months, average temperature of coldest month, average temperature of all months, average temperature of hottest month and maximum temperature of all months) were used in models predicting the phenotype. Where we used these measures as response variables when predicting the temperature within species range (i.e. the environmental temperature to which the species is adapted), we transformed variables with two different formulas to normalize left-shewed distribution (Supplementary Fig. 20).

The temperature measures reflected the full range of global thermal environments occupied by birds. For example, the maximum temperature of all months ranged from −3.8 °C (in the emperor penguin *Aptenodytes forsteri*), through 29.9 °C (median, in the Minas gerais tyrannulet *Phylloscartes roquettei*) to 43.8 °C (in the Basra reed-warbler *Acrocephalus griseldis*). In contrast, the minimum temperature of all months ranged from −35.3 °C (in black-billed capercaillie *Tetrao urogalloides*), through 14.1 °C (median, e.g. in Yellow-breasted apalis *Apalis flavida*) to 24.8 °C (in the Seychelles warbler *Acrocephalus sechellensis*). Notably, our multiple measures of temperature indicated distinct aspects of seasonality in thermal conditions that may require different phenotypic adaptations across avian lineages.

## Phylogeny

To account for evolutionary history, we used 100 fully resolved phylogenetic trees randomly retrieved from the Bird Tree Project[32] in nexus format. First, all analyses were run using single maximum clade credibility tree, prepared with the 'TreeAnnotator' tool implemented in BEAST software (version1.8.0)[65]. Second, all analyses were recalculated with the complete set of 100 phylogenies. To assess the dependence of species traits on phylogeny (shown in Supplementary Table 1), we used the 'phylosig' function implemented in 'phytools' R package (version 1.0-3)[66] to estimate Pagel's $\lambda$[67], varying from 0 (the distribution of the phenotypic trait is completely independent of phylogeny) to 1 (where the distribution of the trait is strongly predicted by the phylogeny).

## Confounding factors

To exclude confounding factors, first we took avian migratory habits available in *AVONET database*[1], which is phylogeny-aggregated version of migration status from *BirdLife International*[35]. Migratory habits were expressed as a categorical variable dividing birds in three distinct classes: resident species (all populations of which reside at single locations through all year), partial migrant species (some populations of which reside at single locations through all year, while others migrate seasonally) and full migrant species (all populations of which migrate seasonally). Second, we assessed geographic range size by aggregating all seasonal ranges[35] of a species[32] to a single polygon and calculated the area (in m²) on an equal-area cylindrical map projection (Eckert IV), ensuring comparable measurements from poles to equator.

## Statistical analyses

All analyses were performed using the R environment version 4.1.1[68]. All hypotheses were tested using phylogenetic linear regression models implemented in the 'phylolm' package (version 2.6.2)[33]. In each of these models we used maximum likelihood lambda settings because they were clearly supported by the model selection procedure. To select the best model we used Akaike Information Criterion (AIC), assuming models with $\Delta$AIC < 7 as somewhat likely and $\Delta$AIC < 2 as most likely. The phylogenetic path analysis was performed using the 'phylopath' R package version 1.1.3[39]. The above package is based on 'phylolm' library[33], thus ensuring the compatibility of our two (non-causal and causal) analytical approaches. To rank the phylogenetic path models, we used the C statistic Information Criterion (CIC), as advised in this kind of analysis[40]. For the exemplary R codes with model specification see Supplementary Note 1. We visualized the results as scatter plots using a custom function based on 'ggplot2' package (version 3.3.6). We worked on the Intel Core i7-7820X CPU, 16-threads of basal speed of 3.60 GHz, using CPU parallel clusters to speed-up analyses.

To predict the phenotype, we used phenotypic traits as response and temperature variables (and absolute latitude for comparison) as predictors. The temperature variables were tightly correlated (Supplementary Table 2), thus in each case we used only one of these variables in a single model. To evaluate Bergmann's rule, we ran univariate models with log-transformed body mass as response and one of the temperature variables as the fixed term. To address Allen's rule, we first built-up null allometric models with log-transformed beak length or tarsus length as responses and log-transformed body mass as principal fixed term to obtain results for the relative (not absolute) size of appendages in all analyses (as Allen's rule states). Then, we extended models by one of temperature variable to formally evaluate simple Allen's rule (independent on trade-off with body size). To test the trade-off hypothesis, we extended the above models by including the interaction term between log-transformed body mass and temperature variable to evaluate the body-size-specific slope of appendage size against temperature, and simultaneously, the temperature-specific slope of appendage size against body size[30].

We additionally extended Bergmann's models (log body mass as response) by including the interaction of temperature and either relative beak or tarsus length. Through this model, we aimed to evaluate the differences in slope of body size against temperature, specific to different appendage lengths to ask whether trade-offs previously seen in Allen's rule (i.e. while explaining the length of appendages) translate also to Bergmann's rule (while explaining body size).

To predict the temperature within species range (representing the 'environmental temperature' for a species), we used transformed temperature variable as response (see Supplementary Fig. 20 for transformation formulas) and phenotypic traits as fixed terms. We did not observe strong correlations between log-transformed body mass, relative beak length, and relative tarsus length (Supplementary Table 3), thus there were no limitations to include them in the same models. We gradually extended the models with the fixed terms of log-transformed body mass (Bergmann's rule) or the relative length of beak or tarsus (both relative to body size, as Allen's rule implies) and each possible combinations of these terms, and then we gradually extended these models with each possible combination of interaction terms to reach the most complex model with three-way interaction between body mass and the relative length of beak and tarsus. The relative lengths of appendages used in this analysis were obtained by extracting phylogenetic residuals of log-log phylogenetic regression of beak or tarsus length against body mass (described as null allometric models in the paragraph above; see Figs. 2a and 3a, by color gradients explained on right). To ensure our results were not sensitive to this particular scaling approach, we did parallel analyses with appendage-size-to-body-size ratios and principal components of phenotype. To assess ratios, we divided the absolute appendage length by body mass and log-transformed the obtained values to normalize their distribution. This approach assumed isometric scaling of body shape with size, but had severe limitations as both ratio of beak length and tarsus length vs body mass were still (negatively) correlated with body mass ($r = -0.95$, $p < 0.001$ and $r = -0.61$, $p < 0.001$, respectively), therefore these results are included only as addendum to our main analyses. Principal components of phenotype were obtained by passing log-transformed and scaled (mean = $0 \pm 1\,SD$) phenotypic traits to 'phyl.pca' function implemented in 'phytools' R package (version 1.0-3)[66]. This approach ensured that body size (PC1) was equally charged by all phenotypic traits, not only by mass.

In the phylogenetic path analysis we built two sets of models integrated from linear regression models described above (excluding those with interactions). First, we gradually integrated all models to predict the phenotype, where we always had log-transformed beak and tarsus length as final responses. These models included: (1) the single direct effect of log-transformed body mass (allometry, the 'null model' because it does not consider temperature) on log beak and tarsus length, (2) the direct effect of allometry and indirect effect of temperature on log body mass (Bergmann's rule), (3) the direct effect of allometry and direct effect of temperature on log beak and tarsus length (Allen's rule) and (4) combination of allometry, Bergmann's and Allen's rule (the 'mixed model'). Second, we gradually integrated all models to predict the temperature within the species range, where we always had (transformed) temperature variable as final response. These models included (1) the single direct effect of log-transformed body mass (allometry) on log beak and tarsus length, (2) the direct effect of allometry and direct effect of log body mass on temperature (Bergmann's rule), (3) the direct effect of allometry and direct effects of log beak and tarsus length on temperature (Allen's rule) and (4) the combination of allometry, Bergmann's and Allen's rule (mixed model).

## Reporting summary

Further information on research design is available in the Nature Portfolio Reporting Summary linked to this article.

## Data availability

The data on the temperature conditions within species geographic ranges generated in this study (Supplementary Data 1) have been deposited in the Dryad repository at https://doi.org/10.5061/dryad. 9ghx3ffn7. The processed polygon layers with avian geographic ranges are available at http://datazone.birdlife.org/[35]. The processed raster layers with monthly temperature conditions are available at https:// worldclim.org/[34]. The phylogenetic trees used in this study are available at https://birdtree.org/[32]. The phenotype data used in this study is available in *AVONET database*[1].

## Code availability

R codes with all analyses (Supplementary Code 1) are deposed in the Dryad repository (https://doi.org/10.5061/dryad.9ghx3ffn7).

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

## Acknowledgements

This study was financially supported by the Polish National Science Centre grant Sonatina 2020/36/C/NZ8/00473. Open access to this article was paid from statutory funds of the Institute of Nature Conservation, Polish Academy of Sciences. MRES was supported by an Australian Research Council grant Discovery Project DP190101244. We thank Kaspar Delhey for discussion and feedback at an earlier stage of this work.

## Author contributions

A.F. designed the study. D.K. and R.M. collected the data. D.K. performed geospatial analyses. A.F. performed statistical analyses. A.F., R.M., and M.R.E.S. interpreted the results. A.F. wrote the first draft of the article. All the authors read and improved the manuscript.

## Competing interests

The authors declare no competing interests.
