## [Peer Review File · Nature Communications]

Allometry reveals trade-offs between Bergmann's and Allen's rules, and different avian adaptive strategies for thermoregulationReviewers' Comments:

Reviewer #1:

Remarks to the Author:

The paper entitled "Allometry reveals trade-offs between Bergmann's and Allen's rules, and different avian adaptive strategies for thermoregulation" reports the findings of phylogenetic comparative analysis of more than 80% of global bird species about how size and/or shape of their body evolved as adaptations to climatic conditions for thermoregulation. Differently from most of previous studies on the topic, which has been largely investigated in various taxa, the manuscript investigated not only the independent variation in body size (Bergmann's rule) and size of unfeathered appendages (tarsus and bill; Allen's rule) according to climate, but also how these traits interact by showing a complex evolutionary scenario. Overall, the main message of the paper is that variation in the relative appendages' length covaries with temperature in a way that depends on body size, and vice versa. For example, larger appendages are found more frequently in species with large body size living in warm temperatures because the large body size constrains heat dissipation, thus favouring alternative strategies. On the other hand, in small species, body size can be sufficient alone to dissipate heat, thus limiting the elongation of the appendages. The size of body and unfeathered appendages therefore resulted as an evolutionary trade-off that reflects distinct, but complementary thermoregulatory strategies.

The study is very interesting, the manuscript is well written, the predictions are clearly stated and the tested hypotheses biologically sounding. Also considering the large taxonomic and geographic coverage of the species included in the analyses, as well as the use of several statistical tests (from phylogenetic linear regressions to path analyses, using many different predictors), my opinion about the paper is positive. However, there are some issues that I would like to raise below to improve, in my opinion, the quality and the readability of the paper.

1. Visualization of the results. The results are comprehensive and quite clear, but I had some problems in visualizing some of the emerging patterns. Indeed, I was rather surprised not to see any figure about the covariation between temperature (annual average or others) and body mass, tarsus length and bill length. I think that it is important for the reader to visualize these crucial relationships. In addition, in Figs. 4 and 5, it is very difficult to interpret the patterns described in the text (i.e. the interactions). It could be thus helpful for the reader if the results of any single slope are presented alone in a single figure showing original data. This could be done in the Supplementary materials, while keeping the actual figures in the main text. In Fig. 1 it is not clear what the word COEFFICIENTS means: are those coefficients the slopes (beta values?) of the relationships between temperature and phenotypic traits? Some clarifications are needed.

2. Data selection.

A. Partial migrants. I could agree about the decision to limit the analyses on resident species only. However, such a choice resulted in the use of mostly tropical species (according to Fig. S5). This is unfortunate because it considerably limits the variability in climatic conditions (and consequently in phenotypic adaptations) used in the analyses. Such a bias in the geographic coverage of the data, and its possible consequences on the results, should be acknowledged in the study. Linked to this, it would be important to know how partial migrants were considered. This is important because many (most?) species breeding at temperate (but also boreal) latitudes are partial migrants, with populations breeding in the northern of the species' range usually overwintering in other regions, but those breeding in the central-southern part of the range being sedentary. How have they been considered? If they were omitted from the analyses, please justify why, and explain how their exclusion might have affected the results.

B. Wide distributed species and within-species variability. The decision to use a single data for each species is more than understandable for the purpose of the study. However, this procedure would not consider the within-species variability in phenotype (but also climatic conditions to which different populations of the same species are exposed). This can be a problem for species showing a large

distribution range (at least continental) because it is well known that they can follow Bergmann's and Allen's rules. In my opinion, additional analyses excluding these widely distributed species would improve the strength of the findings. At least, the authors should mention this problem in the Methods and/or in the Discussion.

3. Residuals vs. ratios. Although residuals are widely used in studies focused on covariation among traits, I am not sure that this is the best way to analyze the present data. I would suggest using ratios rather than residuals, which is recommended especially when the involved variables are strongly correlated, as in the case of the present study according to Fig. 2. The reasons are well-explained in the influential paper by Smith 1999 (*Journal of Human Evolution*, 36(4):423-458). Basically, ratios (with their own issues and caveats) are biological characteristics of different species, while residuals are characteristic of a data set and therefore depend on the species included in the analyses. I would suggest the authors to re-run the analyses using ratios rather than residuals. Maybe, this method can provide more solid estimates, and limit, using the authors' words, "mixtures of expected and ambiguous results" (especially for interactions).

Reviewer #2:

Remarks to the Author:

This manuscript examines global patterns avian body mass and appendage length (beak and tarsus length) as they relate to Bergmann's rule (that animals in warmer climates have smaller body masses) and Allen's rule (that animals in warmer climates have longer appendages). They use an open, previously-published database of avian trait measurements, widely-available range and temperature data, and comprehensive phylogenetic modelling. They find that Allen's rule is more apparent in larger birds (which is entirely sensible) and that bird species in warmer climates tend to have increased either beak length or tarsus length but not necessarily both.

In general, I found this manuscript to be scientifically sound and have only a few small queries and notes. I did, however, find the introduction difficult to follow, and I worry that someone outside the immediate subfield may fail to grasp the importance of this study. If this could be easily fixed, I would give you more concrete suggestions, but unfortunately I will have to be vague in my recommendations for how one might improve the utility of the introduction:

1. A clearer, more concise explanation of what Bergmann's rule and Allen's rule are.
2. A stronger explanation of why one might care about Bergmann's and Allen's rule – is it explaining the phenotypic diversity of life on earth? Understanding the role of climate in shaping macroevolutionary patterns? Making predictions in the face of unprecedented climate change? Something else?
3. A stronger grounding in the previous literature, with a particular focus on what unknown question this manuscript will be addressing. We know a lot about Bergmann's rule in birds, and less about Allen's rule, and what we do know tends to be intraspecific or taxonomically/regionally focused. Why is a global approach needed?

Major comments:

1. The decision to have the response variable of some models be the range temperature is slightly discomfiting – obviously this is the most straightforward way to test for trade-offs between the two appendages in the Allen's rule models, but conceptually readers may find it difficult. Stressing the biological interpretation of adaptations for particular climate rather than simply reporting the stats might help.
2. In general, I found the reporting of the results to be so focused on the model details that the biological context was sometimes lost – occasionally reminding the reader (who, in a general journal like *Nature Communications*, is not likely to be an expert in this field and/or this methodology) what

you're testing, why, and how to interpret the results would go a long way to improving the readability of this.

Minor comments:

1. Figures: Even forewarned by the disclaimer that the figures were not up to Nature Communications standards, I had a dreadful time reading the figures – they were rendered very poorly in the reviewer copy. Please be aware of this for future submissions!
2. Having Figure 1 be a summary of model outputs (which only specialists can understand) and Figure 2 demonstrating that body mass is linked to both beak length and tarsus length (which is fairly obvious) is perhaps not the banner headline you want, for a journal like Nature Communications. Researchers outside the subfield, I think, will want to see temperature plotted against mass and then temperature against the extremity-mass residual – show us what Bergmann's rule and Allen's rule actually looks like! (I'd also love a map. I know they're a pain to do. But that would be a lovely visualisation of what's going on here, one that non-specialists could understand.)
3. L158 – the phrasing of "Let's" strikes me as too informal for a scientific manuscript. Even rephrasing as "For example, consider a species experiencing a temporal increase..." would improve this, in my opinion.
4. L218 – Could the pattern of shortening tarsi toward the equator be potentially driven by hummingbirds?
5. Figure 5 – how interpretable is this figure to those with red-green colourblindness? I think the figure itself is probably fine, but the use of the words "red", "green", and "blue" in the caption might be confusing. (For that matter, I'm not entirely sure I understand what that sentence means.)
6. L367-368 – I don't follow this conclusion (that body size is constrained by the size of the appendages), though it's entirely possible that I missed a detail somewhere. What is your evidence for this statement? Can you make this line of reasoning more pronounced in your discussion? (If this is indeed well-supported by your results, that's fascinating.)
7. L390 – Along the lines of one of my major comments, I'm not following why one would expect a high correlation between range temperature and physical phenotype. Can you expand on this? Or just rephrase? Something along the lines of, for example, "It is worthwhile emphasizing that many other elements combine to help birds meet their thermoregulatory requirements, e.g. through variation in ...behavior, a fact perhaps partially reflected in the low correlation between ambient temperature and physical phenotype in even our strongest models ($R^2 = 0.20$)."
8. L404-406 – I'm not entirely sure what you mean here, but regardless, I'm not sure it's the first, as other studies have demonstrated simultaneous applications of Bergmann's and Allen's rules in e.g. shorebirds (McQueen et al. 2022 Nature Communications), or Allen's rule in e.g. rodents (Alhajeri et al. 2020 GEB).
9. L468: I think there's a typo somewhere in here. $\Delta\text{-AIC} < 2$ being 'substantially better', perhaps?

Highlight explanation:

Reviewers comments

Authors response

Author's preface

The manuscript has been substantially changed and it requires some general explanation and justification of our modifications.

We agree with the Reviewer 1 that the inclusion of only resident species in our analyses was a limitation of this study. Therefore, we have now extended the analyses to (almost) all avian species (N = 9962, 99.7% of all bird species, for which geographic data was available). It is now including residents (N = 7936), partial-migrants (N = 1109), and full-migrants (N = 917). Note that these migration categories are based on "migration status" (here named "migratory habits") from the Birdlife database (rearranged by AVONET) and the present number of resident species is not equal to the previous sample size. This is because in previous approach we did not use "migration status", but based our choice of species on geographic-range vector layer, taking species with only year-round ranges (excluded those containing at least one polygon assigned as the presence in breeding or wintering period). Some of the resident species formerly included (N = 561) were indeed partial migrants, if we look at „migration status". This means that in some species previously included as resident, some populations migrate seasonally.

The inclusion of all birds to analyses revealed almost identical patterns. Note please that for technical reasons, we changed the main temperature variable (presented in main article, not in supplements) into maximum temperature of all months, as this (likewise min temp all months) is more reliable when assessing temperature in residents and migrants (unlike average temperature all months that would be a large simplification in the case of migrating species). To compare the consistency of results before and after, please compare the previous models (e.g. the most complex model with three-way interaction) with the present one (Fig. S6c), which is now in supplementary materials. These results show both qualitative and quantitative consistency. Note also that the models on resident species are still shown (upper rows of Fig. S8, S9 and S10), but as a model nested within migratory habits as three-level categorical variable.

Taking into account the suggestion of the Reviewer 2, we have decided to change the coloration of figures through the entire paper. We used unique color gradients for all phenotypic traits (and distinct for temperature) to avoid confusions when all analyzed traits are marked similarly. Moreover, this is better for the interpretation of models with three-way interactions (new coloration is based on unique scheme, shown in Fig. 4, legend in bottom-right).

REVIEWER COMMENTS

Reviewer #1 (Remarks to the Author):

The paper entitled "Allometry reveals trade-offs between Bergmann's and Allen's rules, and different avian adaptive strategies for thermoregulation" reports the findings of phylogenetic comparative analysis of more than 80% of global bird species about how size and/or shape of their body evolved as adaptations to climatic conditions for thermoregulation. Differently

from most of previous studies on the topic, which has been largely investigated in various taxa, the manuscript investigated not only the independent variation in body size (Bergmann's rule) and size of unfeathered appendages (tarsus and bill; Allen's rule) according to climate, but also how these traits interact by showing a complex evolutionary scenario. Overall, the main message of the paper is that variation in the relative appendages' length covaries with temperature in a way that depends on body size, and vice versa. For example, larger appendages are found more frequently in species with large body size living in warm temperatures because the large body size constrains heat dissipation, thus favouring alternative strategies. On the other hand, in small species, body size can be sufficient alone to dissipate heat, thus limiting the elongation of the appendages. The size of body and unfeathered appendages therefore resulted as an evolutionary trade-off that reflects distinct, but complementary thermoregulatory strategies.

The study is very interesting, the manuscript is well written, the predictions are clearly stated and the tested hypotheses biologically sounding. Also considering the large taxonomic and geographic coverage of the species included in the analyses, as well as the use of several statistical tests (from phylogenetic linear regressions to path analyses, using many different predictors), my opinion about the paper is positive. However, there are some issues that I would like to raise below to improve, in my opinion, the quality and the readability of the paper.

1. Visualization of the results.

The results are comprehensive and quite clear, but I had some problems in visualizing some of the emerging patterns. Indeed, I was rather surprised not to see any figure about the covariation between temperature (annual average or others) and body mass, tarsus length and bill length. I think that it is important for the reader to visualize these crucial relationships. In addition, in Figs. 4 and 5, it is very difficult to interpret the patterns described in the text (i.e. the interactions). It could be thus helpful for the reader if the results of any single slope are presented alone in a single figure showing original data. This could be done in the Supplementary materials, while keeping the actual figures in the main text. In Fig. 1 it is not clear what the word COEFFICIENTS means: are those coefficients the slopes (beta values?) of the relationships between temperature and phenotypic traits? Some clarifications are needed.

Thank you for this comment. Your opinion about visualization largely converges with the second Reviewer, who gave us several tips how to improve the presentation of results in this study. Therefore, visualization of the results is now changed. Addressing your questions in details:

- a. "I was rather surprised not to see any figure about the covariation between temperature (annual average or others) and body mass, tarsus length and bill length". We added the simple Bergmann's and Allen's rule slopes in a Fig. 1c, 2d, 3d, S1, S2a and S3a.
- b. "In Figs. 4 and 5, it is very difficult to interpret the patterns described in the text (i.e. the interactions). It could be thus helpful for the reader if the results of any single slope are presented alone in a single figure showing original data. This could be done in the Supplementary materials, while keeping the actual figures in the main text". We wish to present the relationships on single plots with sole trends, as you advised. This is however very tricky, if we realize that it multiplies the number of plots. For example, the only exemplar relationships presented in the main text in Figs. 2e, 3e or 4c-f, could be presented as 200 distinct plots (instead of eight presented currently). Such a large number of plots may produce adverse outcomes – confuse the readers

rather than convince them. So, we are very sorry to refuse your suggestion. Apart from the new coloration that is now unified through the paper, we still keep the same method to plot interactions. However, we agree with you that Fig. 4f (formerly, Fig. 5) with three-way interaction is the most complex and requires large focus to be understood. We therefore simplified it (partly according to your comment), by presenting relationships between y and x_1 across x_2 (as in other figures with two-way interactions), while presenting this relationship across factor x_3 in rows (Fig. S5). We hope this solution, at least partially, satisfy your concerns.

c. "In Fig. 1 it is not clear what the word COEFFICIENTS means".

Not applicable any more, as now this figure is replaced by scatterplots.

2. Data selection.

A. Partial migrants. I could agree about the decision to limit the analyses on resident species only. However, such a choice resulted in the use of mostly tropical species (according to Fig. S5). This is unfortunate because it considerably limits the variability in climatic conditions (and consequently in phenotypic adaptations) used in the analyses. Such a bias in the geographic coverage of the data, and its possible consequences on the results, should be acknowledged in the study. Linked to this, it would be important to know how partial migrants were considered. This is important because many (most?) species breeding at temperate (but also boreal) latitudes are partial migrants, with populations breeding in the northern of the species' range usually overwintering in other regions, but those breeding in the central-southern part of the range being sedentary. How have they been considered? If they were omitted from the analyses, please justify why, and explain how their exclusion might have affected the results.

We think this is very valid criticism, and hence we now include all birds in the analysis (see the preface), allowing us to assess whether and how partial-migrants (and full-migrants too) confound our results. We accounted for migratory habits by including it as fixed term in our main models (Fig. S7), showing that the observed patterns remain undisturbed. Also, we tested for the interaction of our focal models with migration category (Fig. S8-S10), to see whether and how trade-offs vary between resident partial- and fully-migrating species. It turns out that the discovered patterns are not specific to only resident species, but are prominent in partial migrants and (to some extent) in full migrants; see changes in the section "Excluding possible confounding factors" and methods.

B. Wide distributed species and within-species variability. The decision to use a single data for each species is more than understandable for the purpose of the study. However, this procedure would not consider the within-species variability in phenotype (but also climatic conditions to which different populations of the same species are exposed). This can be a problem for species showing a large distribution range (at least continental) because it is well known that they can follow Bergmann's and Allen's rules. In my opinion, additional analyses excluding these widely distributed species would improve the strength of the findings. At least, the authors should mention this problem in the Methods and/or in the Discussion.

Thank you for these comments. Within-species variability (increasing with range size) could really be an issue and we are happy to address it. However, exclusion of these "widely distributed species" requires us to make a very subjective decision - which species are widely distributed? What is a threshold of range size that makes species widely distributed?. We thus decided to extend our models by variable "species range size", either as fixed (Fig. S11) and interaction term (Fig. S12-S14). This allowed us to address the problem more conscientiously, rather than subjectively limiting the sample size to some species. It revealed that the trade-offs hold when controlling for range size (Fig. S11) and that the trade-offs are

very similar across geographic range sizes (Fig. S12-S14). Importantly, an interaction with range size is non-significant or has a little quantitative (not qualitative) influence on the results. For more information, see changes in the section “Excluding possible confounding factors” and methods.

3. Residuals vs. ratios. Although residuals are widely used in studies focused on covariation among traits, I am not sure that this is the best way to analyze the present data. I would suggest using ratios rather than residuals, which is recommended especially when the involved variables are strongly correlated, as in the case of the present study according to Fig. 2. The reasons are well-explained in the influential paper by Smith 1999 (*Journal of Human Evolution*, 36(4):423-458). Basically, ratios (with their own issues and caveats) are biological characteristics of different species, while residuals are characteristic of a data set and therefore depend on the species included in the analyses. I would suggest the authors to re-run the analyses using ratios rather than residuals. Maybe, this method can provide more solid estimates, and limit, using the authors’ words, “mixtures of expected and ambiguous results” (especially for interactions).

Thank you for this comment. However, we disagree that ratios are the best approach to our problem. Ratios of x_1/x_2 are fine when the relationship between x_1 and x_2 is linear, but this is not the case of appendage length and body size across avian species; those are usually logarithmic relationships (see section “Allometry of appendages”). We calculated the ratios of appendage length vs body mass, and the obtained values are still heavily (but negatively) correlated with body mass (beak, $r = -0.95$; tarsus, $r = -0.61$; see figure below, and changes in section “Statistical analyses”); this is the case when we calculated these ratios on log-transformed and raw values and when we log-transformed ratio values.

The high negative correlation means that inclusion of above ratios as a predictors in our models will cause these models to detect inverse Bergmann’s rule (not Allen’s rule that assumes the relative changes in appendages). Furthermore, inclusion of highly correlated covariates in one model is not advised.

Addressing your argument on dataset-specific character of residuals: It is true, but this problem is partly solved by including phylogeny as a random effect (as in our approach applying phylogenetic linear regression). That said, it has little influence on the values of relative size of appendages in our case; current residuals (assessed on all avian species) are both tightly correlated with previous residuals (assessed on dataset with resident species only): $r = 0.99$ for beak and $r = 0.99$ for tarsus, respectively.

Nevertheless, extending our analyses by alternative models with ratios align with our multi-analytical approach to the entire problem, so we are happy to include it as additional evidence for the trade-offs between Allen's and Bergmann's rules (see Fig. S15). So, following your comment we show that the model with three-way interaction between: a) ratio of beak length to body mass, b) ratio of tarsus length to body mass and c) log body mass, is the best supported model among those explaining the temperature within species ranges (Fig. S15). See also the changes in the section "Excluding possible confounding factors" and methods.

Reviewer #2 (Remarks to the Author):

This manuscript examines global patterns avian body mass and appendage length (beak and tarsus length) as they relate to Bergmann's rule (that animals in warmer climates have smaller body masses) and Allen's rule (that animals in warmer climates have longer appendages). They use an open, previously-published database of avian trait measurements, widely-available range and temperature data, and comprehensive phylogenetic modelling. They find that Allen's rule is more apparent in larger birds (which is entirely sensible) and that bird species in warmer climates tend to have increased either beak length or tarsus length but not necessarily both.

In general, I found this manuscript to be scientifically sound and have only a few small queries and notes. I did, however, find the introduction difficult to follow, and I worry that someone outside the immediate subfield may fail to grasp the importance of this study. If this could be easily fixed, I would give you more concrete suggestions, but unfortunately I will have to be vague in my recommendations for how one might improve the utility of the introduction:

1. A clearer, more concise explanation of what Bergmann's rule and Allen's rule are.
2. A stronger explanation of why one might care about Bergmann's and Allen's rule – is it explaining the phenotypic diversity of life on earth? Understanding the role of climate in shaping macroevolutionary patterns? Making predictions in the face of unprecedented climate change? Something else?
3. A stronger grounding in the previous literature, with a particular focus on what unknown question this manuscript will be addressing. We know a lot about Bergmann's rule in birds, and less about Allen's rule, and what we do know tends to be intraspecific or taxonomically/regionally focused. Why is a global approach needed?

Points 1-3. Thank you for these suggestions. We asked some non-experts for their thoughts on the paper and some of them indeed agreed that a lot of questions arose after a talk that one of us presented on this study. As a response to these comments, we extended the introduction with more clear, concise and stronger (we hope) explanation of Bergmann's and Allen's rules, implementing your suggestions about macro-evolutionary patterns in phenotype as a consequence of climate change (see changes in first paragraph of introduction, that is now spread to two and the last sentence of conclusions). We also have more strongly embedded our study in the context of current knowledge (see new paragraph 2. of introduction).

We also took the general idea of this comment and ran ahead by including some examples of a species with large/small and median body size, relative length of both appendages (see newly added sentences to section of methods "phenotype") and temperatures (see section of methods "Temperature") to provide a better conceptual image for the potential readers who are not familiar with global biogeographic diversity of birds.

Major comments:

1. The decision to have the response variable of some models be the range temperature is slightly discomfiting – obviously this is the most straightforward way to test for trade-offs between the two appendages in the Allen's rule models, but conceptually readers may find it

difficult. Stressing the biological interpretation of adaptations for particular climate rather than simply reporting the stats might help.

We agree, so we changed the text of section “Bergmann’s and Allen’s rules in climatic adaptations” to be focused more on interpretation of patterns rather than raw statistics. We also introduced minor changes to whole article, explaining that these models indicate temperature preferences or occurrences in given climates, rather than increase in temperature being driven by appendage and body size, as might have been misleadingly inferred from the text. This is very relevant comment, thank you!

2. In general, I found the reporting of the results to be so focused on the model details that the biological context was sometimes lost – occasionally reminding the reader (who, in a general journal like Nature Communications, is not likely to be an expert in this field and/or this methodology) what you’re testing, why, and how to interpret the results would go a long way to improving the readability of this.

We agree. We simplified the results, by avoiding the discussion on the stats (partly addressed in the answer to your comment above). However, at some points we could not reduce the very necessary statements, which we think are reasonable. We hope that this is what you expected.

Minor comments:

1. Figures: Even forewarned by the disclaimer that the figures were not up to Nature Communications standards, I had a dreadful time reading the figures – they were rendered very poorly in the reviewer copy. Please be aware of this for future submissions!

We apologize for this. Now, in addition to rasterized figures nested in the text document, we attached original figures (in vector graphic) as separate PDF files.

2. Having Figure 1 be a summary of model outputs (which only specialists can understand) and Figure 2 demonstrating that body mass is linked to both beak length and tarsus length (which is fairly obvious) is perhaps not the banner headline you want, for a journal like Nature Communications. Researchers outside the subfield, I think, will want to see temperature plotted against mass and then temperature against the extremity-mass residual – show us what Bergmann’s rule and Allen’s rule actually looks like! (I’d also love a map. I know they’re a pain to do. But that would be a lovely visualisation of what’s going on here, one that non-specialists could understand.)

- a. We realize that Figure 1 may be difficult to be understood and non-expert audience may expect something different. We thus removed Fig. 1 and included the corresponding results in the new Fig. 1, 2 and 3 which present scatter plots instead of effect sizes (and which addresses also your other comments, see below).
- b. We disagree that previous Fig. 2 (allometry of appendages) is unnecessarily shown. We feel that it is crucial for the whole study, since: 1) these allometric models are null (reference) models in selection procedures while explaining the length of appendages; 2) relative lengths of appendages (which are products/residuals of these allometric models) appear throughout the paper; 3) changes in allometry across temperature gradient (the extension of allometric models in next figures) is an important part of our conclusions. To sum up, we stand at the position to keep the allometric relationships in the paper and we include it now as Fig. 2a and 3a. We

added coloration to the data points to illustrate what residuals (relative length of appendages) mean, which may be helpful for brief understanding of statistical procedures.

- c. We have visualized Bergmann's and Allen's rules as scatter plots (Fig. 1c, 2d and 3d), as you suggested.
- d. We also followed your suggestion on the addition of the maps (Fig. 1a, 2b, 3b and 4a). We think this indeed helps to understand the data and nature of our analysis, as well as the avian climatic and phenotypic variation in respect to geography.

3. L158 – the phrasing of “Let's” strikes me as too informal for a scientific manuscript. Even rephrasing as “For example, consider a species experiencing a temporal increase...” would improve this, in my opinion.

We changed this sentence, as you suggested.

4. L218 – Could the pattern of shortening tarsi toward the equator be potentially driven by hummingbirds?

Shortening tarsi toward hot climates is very prominent in hummingbirds. However, we show (**figure below**), that within the set of all avian species except hummingbirds, the pattern still holds. It is weaker, but that is not surprising, since hummingbirds are the smallest birds and introduce substantial additional variation to our study system. Therefore, our results seem to be universal, as they apply to other small birds. We recognize that this comment is more raised out of curiosity, rather than a specific suggestion. We are happy to address it in these response comments, however, we have opted not to present any analyses within subsets of particularly taxa. The paper is focused on global avian diversity, and we don't want to complicate the narration further by dwelling deeper into some taxonomic groups.

5. Figure 5 – how interpretable is this figure to those with red-green colourblindness? I think the figure itself is probably fine, but the use of the words “red”, “green”, and “blue” in the caption might be confusing. (For that matter, I’m not entirely sure I understand what that sentence means.)

You are right. In fact, the colors are only helpful, not crucial, to understand the figure. Although this is a very minor detail, we provided uniform coloration across the paper and explained it in legends in each figure. Entire scheme of coloration is explained in legend in Fig. 4 (bottom-right). We investigated the figures with colorblindness simulator (<https://www.color-blindness.com/coblis-color-blindness-simulator/>). Two-way interactions are readable even for those with monochromacy/achromatopsia (there are still differences in color brightness than you may interpret by pictograms). The problem may be for plots with three-way interactions (as in Fig. 4f) – however, we provided alternative visualization of this model by following two-way interaction schemes interpretable by color brightness (see Fig. S5).

6. L367-368 – I don’t follow this conclusion (that body size is constrained by the size of the appendages), though it’s entirely possible that I missed a detail somewhere. What is your evidence for this statement? Can you make this line of reasoning more pronounced in your discussion? (If this is indeed well-supported by your results, that’s fascinating.)

This is true, we actually did not adequately demonstrate this in the original version of the paper. So, we have added some additional analyses to test this; see changes in “Allen’s vs

Bergmann's rule in allometry", new Fig.S4 and explanation of this approach in the methods. It turns out that body size is really constrained by the size of the appendages across temperature in the expected way, so the statement being the subject of this comment is still valid.

7. L390 – Along the lines of one of my major comments, I'm not following why one would expect a high correlation between range temperature and physical phenotype. Can you expand on this? Or just rephrase? Something along the lines of, "It is worthwhile emphasizing that many other elements combine to help birds meet their thermoregulatory requirements, e.g. through variation in ...behavior, a fact perhaps partially reflected in the low correlation between ambient temperature and physical phenotype in even our strongest models ($R^2 = 0.20$)."

We meant that the variation in temperature preferences is not only attributed to body size and shape, but also other (non-addressed here) elements, possibly related also with physiological properties. We rephrased it with your tip and now it should be clearer, thank you.

8. L404-406 – I'm not entirely sure what you mean here, but regardless, I'm not sure it's the first, as other studies have demonstrated simultaneous applications of Bergmann's and Allen's rules in e.g. shorebirds (McQueen et al. 2022 Nature Communications), or Allen's rule in e.g. rodents (Alhajeri et al. 2020 GEB).

We changed this sentence to be more precise. We hope our message is clear now. Thank you for the new reference.

9. L468: I think there's a typo somewhere in here. Delta-AIC < 2 being 'substantially better', perhaps?

Yes, thank you for the notice. We've corrected it.

Reviewers' Comments:

Reviewer #1:

Remarks to the Author:

The Authors have satisfactorily addressed most of the points I raised in the first review, and I think that the conclusions of the paper are now even stronger and clearer. There still is a small number of issues left to be solved, but I think they should be easy to fix. Please, see my minor specific comments below (unfortunately I cannot find page and line numbers, so I report the page number and the sentence of each comment)

- 1) The quality of some of the figures is very poor to properly understand their meaning. This is especially the case for those including multiple panels reported in the Supplementary Materials.
- 2) P4. "Indeed, Allen explicitly embedded his rule in the context of Bergmann's rule, proposing a scenario where animals could maintain desired body size (or even show converse Bergmann's patterns) under increasing temperatures by manipulating the size of appendages". I do not like this sentence because it seems that animals consciously act to change their phenotypic traits. Animals adapt to characteristics of the environment where they live. I would suggest rephrasing.
- 3) P5. "We then investigate how body size and appendage length interact with each other to predict the preferred temperature of the species range to ask how the phenotype adapts to different climates". I do not really understand what the authors mean with PREFERRED TEMPERATURE. I would suggest using a different terminology throughout the text.
- 4) Figure 4f. I still have problems in visualizing the datapoints producing these very different trends. In the current version, there is an agglomeration of dots, and it is almost impossible to discern which ones generate each trend shown by coloured lines. I understand that the huge sample size may prevent a clear visualization, but I suggest to re-think (again, sorry) the way how this result is presented. Figure S5 is much clearer but, in its current version, it is impossible to be properly read given the very poor quality of the image.
- 5) P25. "We also retrieved absolute latitude from the centroids of the species ranges". Some more details should be given for partial- and full-migrants.
- 6) Discussion. Excluding possible confounding factors. This is a more than welcome section that it is well integrated in the Discussion, as it reports the interpretation of novel analyses about migratory behaviour and distribution range. However, I think that this section would also benefit from a paragraph describing some very important ecological factors (shortly mentioned in the very last part of the Conclusion section) that may have constrained (or favoured) variation in body size and appendage length, and were not treated (the authors did not have to do) in the analyses. Among them, I think that it is important to mention, at least, trophic level and foraging strategy adopted, as they can be tightly linked with both body size and bill shape.

Reviewer #2:

Remarks to the Author:

This revised manuscript demonstrates that global interspecific variation in avian size and shape is structured in part by complementary adherence to Bergmann's and Allen's rules (two classic ecogeographic thermoregulatory phenomena). The current draft represents an incredible amount of work and what seems to be an impressive attempt to address the reviewers' comments. (Though for what it's worth I agree with Reviewer 1 that the inter-specific variation in climatic niche breadth is likely confounding these analyses.)

This study seems to be technically sound, and I find this version much more readable. I only have a few outstanding comments, all minor and mostly superficial.

Page 3, first full paragraph – the first sentence ('for the last 150 years') is contradicted later in the paragraph ('However, we are far from'). You can be much more clear/explicit about the novel contribution of this study to the literature: namely, that it is generally unknown how Bergmann's rule and Allen's rule apply **simultaneously**.

Page 12, first full paragraph – missing parenthesis after "Fig 3b"

Page 23, final line – missing word, "has **an** almost invisible beak"

Page 24, second line – missing word, "in which **the** beak reaches...of **the** body length" (I'd also recommend phrasing along the lines of "the total body length" or "the beak accounts for approximately", to make it clear that the body length here includes the bill)

Page 25, bottom of page – I'm a little baffled by this "minimum temperature of all months" trait; surely one of the Antarctic penguins has a lower minimum range temperature than a black-billed capercaillie? What am I missing about this variable?

Page 58 – specify what these p-values are measuring. Is it that the values of lambda are significantly different from 1? From 0? From something else?

Highlight explanation:

Reviewers comments

Authors response

REVIEWERS' COMMENTS

Reviewer #1 (Remarks to the Author):

The Authors have satisfactorily addressed most of the points I raised in the first review, and I think that the conclusions of the paper are now even stronger and clearer. There still is a small number of issues left to be solved, but I think they should be easy to fix. Please, see my minor specific comments below (unfortunately I cannot find page and line numbers, so I report the page number and the sentence of each comment)

1) The quality of some of the figures is very poor to properly understand their meaning. This is especially the case for those including multiple panels reported in the Supplementary Materials.

We are very sorry that you were not satisfied with the quality of display items. However, please note that this was due to a high level of compression that occurred at the submission process. We have provided high-quality figures in a vector graphic format (svg or pdf files) packed in zip/rar archives, but it's possible that you did not received it. We assume that this will not be an issue during production.

2) P4. "Indeed, Allen explicitly embedded his rule in the context of Bergmann's rule, proposing a scenario where animals could maintain desired body size (or even show converse Bergmann's patterns) under increasing temperatures by manipulating the size of appendages". I do not like this sentence because it seems that animals consciously act to change their phenotypic traits. Animals adapt to characteristics of the environment where they live. I would suggest rephrasing.

This may be indeed interpreted like you suggested. We therefore changed the sentence to a "Indeed, Allen explicitly embedded his rule in the context of Bergmann's rule, proposing a scenario where animals **lineages** could maintain **desired their optimum** body size (or even show converse Bergmann's patterns) under increasing temperatures **by manipulating through changes in** the size of appendages **over evolutionary time**". We hope these minor changes would help to understand that the sentence is about evolutionary process but not a conscious animal decision.

3) P5. "We then investigate how body size and appendage length interact with each other to predict the preferred temperature of the species range to ask how the phenotype adapts to different climates". I do not really understand what the authors mean with PREFERRED TEMPERATURE. I would suggest using a different terminology throughout the text.

Thank you for this note. We missed some words and hence produced a misunderstanding, sorry. However, we feel that the term "preferred temperature" is very intuitive and helps to

understand the interpretation of models with temperature as response. It was a reply to the second reviewer's comment and it was appreciated. We therefore changed this sentence to "We then investigate how body size and appendage length interact with each other to predict the preferred temperature **experienced by** of the species **within its geographic** range ("**preferred temperature**" hereafter, **as this value reflects the temperature to which that species is adapted**) to ask how the phenotype adapts to different climates". We hope this clarification at the first mention of the term will help to understand the discussion on "temperature preferences" and therefore the main conclusions of our modeling procedure.

4) Figure 4f. I still have problems in visualizing the datapoints producing these very different trends. In the current version, there is an agglomeration of dots, and it is almost impossible to discern which ones generate each trend shown by coloured lines. I understand that the huge sample size may prevent a clear visualization, but I suggest to re-think (again, sorry) the way how this result is presented. Figure S5 is much clearer but, in its current version, it is impossible to be properly read given the very poor quality of the image.

Again, we are sorry that you were not satisfied with display items. As in the comment above, this was possibly due to the loss in figure quality after a compression. We agree that these plots (even in a high quality resolution) are not easy, but it is not surprising that models involving four factors (temperature, body size, beak length and tarsus length) are difficult to interpret. Therefore, we decided to present plots as in Fig. 4f only to briefly show what is going on with different combinations of phenotypic traits and to show that these patterns are very complex. As you note, Fig. S5 provides more detail (and is more readable when in original quality). However, the current plotting approach allows for quick comparison between the patterns inferred from Fig. 4f and other figures (Fig. S6, S7c, S10, S11c and S14) and enable us to visualise more readily the key patterns, excluding the influence of confounding factors. This would be much more difficult (and space- and time-consuming) when all of these figures were plotted in the way as in Fig. S5. Therefore, we would prefer to hold with the current presentation approach.

5) P25. "We also retrieved absolute latitude from the centroids of the species ranges". Some more details should be given for partial- and full-migrants.

This was mentioned in the earlier sentence: "Fifth, we took all (breeding, winter and year-round) species ranges...". But as you suggested, we clarified it and added a comment on how this worked for residents versus migrants: "We also retrieved absolute latitude from the centroids of the **above species ranges (breeding, winter and year-round, summarized to a single polygon per species), which described a simple geographic variation across species.**"

6) Discussion. Excluding possible confounding factors. This is a more than welcome section that it is well integrated in the Discussion, as it reports the interpretation of novel analyses about migratory behaviour and distribution range. However, I think that this section would also benefit from a paragraph describing some very important ecological factors (shortly mentioned in the very last part of the Conclusion section) that may have constrained (or favoured) variation in body size and appendage length, and were not treated (the authors did not have to do) in the

analyses. Among them, I think that it is important to mention, at least, trophic level and foraging strategy adopted, as they can be tightly linked with both body size and bill shape.

You are right. Following your tip, we included a short, three-sentence paragraph on this issue:

“Notably, there is a wider list of important ecological factors constraining or favoring variation in body size and shape, e.g. trophic levels or foraging techniques, although they are also themselves constrained by phylogeny to some extent, which we control for. Nevertheless, we believe that it is likely that these constraints influenced (or were influenced by) the Bergmann-Allen trade-off. Understanding of this issue would benefit from a deeper dive into the relationship between climatic, phenotypic and ecological variation across animals.”

Reviewer #2 (Remarks to the Author):

This revised manuscript demonstrates that global interspecific variation in avian size and shape is structured in part by complementary adherence to Bergmann's and Allen's rules (two classic ecogeographic thermoregulatory phenomena). The current draft represents an incredible amount of work and what seems to be an impressive attempt to address the reviewers' comments. (Though for what it's worth I agree with Reviewer 1 that the inter-specific variation in climatic niche breadth is likely confounding these analyses.)

This study seems to be technically sound, and I find this version much more readable. I only have a few outstanding comments, all minor and mostly superficial.

Page 3, first full paragraph – the first sentence ('for the last 150 years') is contradicted later in the paragraph ('However, we are far from'). You can be much more clear/explicit about the novel contribution of this study to the literature: namely, that it is generally unknown how Bergmann's rule and Allen's rule apply *simultaneously*.

Thank you for this note, we added the alternative "how they evolve simultaneously" in the parenthesis after original statement.

Page 12, first full paragraph – missing parenthesis after "Fig 3b"

We inserted the lacking parenthesis, thank you

Page 23, final line – missing word, "has *an* almost invisible beak"

We inserted the lacking "an", thank you

Page 24, second line – missing word, "in which *the* beak reaches...of *the* body length" (I'd also recommend phrasing along the lines of "the total body length" or "the beak accounts for approximately", to make it clear that the body length here includes the bill)

Thank you for this correction. We changed the phrase to "the beak accounts for approximately the 50% of the body length", as you suggested

Page 25, bottom of page – I'm a little baffled by this "minimum temperature of all months" trait; surely one of the Antarctic penguins has a lower minimum range temperature than a black-billed capercaillie? What am I missing about this variable?

Yes, it is true. It turns out that these penguins breed in the coastal Antarctic and overwinter on islands closer to the equator. The seasonal migration and the oceanic climate means this species experiences very cold temperatures in breeding periods, but milder winters than it might be expected (this is not an error driven by inverse seasonalities between southern and northern hemispheres). Capercaillie resides all year in a continental climate (which is very variable annually), hence experiences relatively hot summers and cold winters. This is not only the case

of capercaillie, but many other continental species. However, Antarctic penguins are still ranked 49th (out of 9962) in terms of avian species experiencing the harshest winters (this still makes his climate very harsh!).

Page 58 – specify what these p-values are measuring. Is it that the values of lambda are significantly different from 1? From 0? From something else?

The p values indicates whether the lambda is significantly different from 0. We clarified this in the caption of Tab. 1.